**Subject Area:**
developmental biology/cellular biology

cell fate commitment, *Drosophila melanogaster*, stem cell lineages, amplifier, poising, timeliness

**Author for correspondence:**
Yan Song
e-mail: yan.song@pku.edu.cn

# Faster, higher, stronger: timely and robust cell fate/identity commitment in stem cell lineages

Kun Liu[1,2], Ke Xu[1] and Yan Song[1,2]

[1]Ministry of Education Key Laboratory of Cell Proliferation and Differentiation, School of Life Sciences, and
[2]Peking-Tsinghua Center for Life Sciences, Peking University, Beijing 100871, People's Republic of China

(iD) YS, 0000-0002-1413-6123

Precise specification of cell fate or identity within stem cell lineages is critical for ensuring correct stem cell lineage progression and tissue homeostasis. Failure to specify cell fate or identity in a timely and robust manner can result in developmental abnormalities and diseases such as cancer. However, the molecular basis of timely cell fate/identity specification is only beginning to be understood. In this review, we discuss key regulatory strategies employed in cell fate specification and highlight recent results revealing how timely and robust cell fate/identity commitment is achieved through transcriptional control.

## 1. Introduction

### 1.1. Cell fate and bistability

> 'Timeliness is best in all matters.'
> Hesiod

Cell fate decision-making in stem cell lineages is often binary: that is, newly born sibling cells ultimately rest in two discrete, steady and switch-like equilibrium states [1–3]. Such bistable cell fate specification can be achieved by either intercellular or intracellular signalling network [3,4]. Lateral inhibition is a typical intercellular competition that diversifies cell fates: two adjacent sibling cells inhibit each other, eventually resulting in a winner cell and a loser cell that adopts mutually exclusive cell fates (figure 1). A classic example is Notch-Delta-mediated lateral inhibition, which has been implicated in cell fate determination within various stem cell lineages [5,6]. On the other hand, intracellular bistable regulation converts differential external or intrinsic signals into binary cell fates [7–15] (figure 1). For example, in *Drosophila* ovarian germline stem cell (GSC) lineages, differential strengths of niche-derived BMP signal can be sensed by a GSC and its sibling cystoblast cell, and converted into bistable cell fates through cell-autonomous negative feedback loops [16,17]. Another example is asymmetric segregation of Numb protein during mitosis dictates neural stem cell versus neural progenitor/precursor binary cell fate decisions [18–23]. Similar strategies are employed during cell fate decision-making in a wide range of stem cell lineages [7], including the very first cell fate differentiation events in early mammalian development [24,25].

The intercellular and intracellular regulatory strategies are not mutually exclusive. Instead, they are complementary to each other and can be used in combination to confer differential cell fates in a precise and robust manner.

### 1.2. The beginning of the end—from cell fate decision to cell fate commitment

The past decade has witnessed an extraordinary progress in understanding asymmetric stem cell division, in which the intrinsic cell polarity cues and

royalsocietypublishing.org/journal/rsob    Open Biol. **9**: 180243

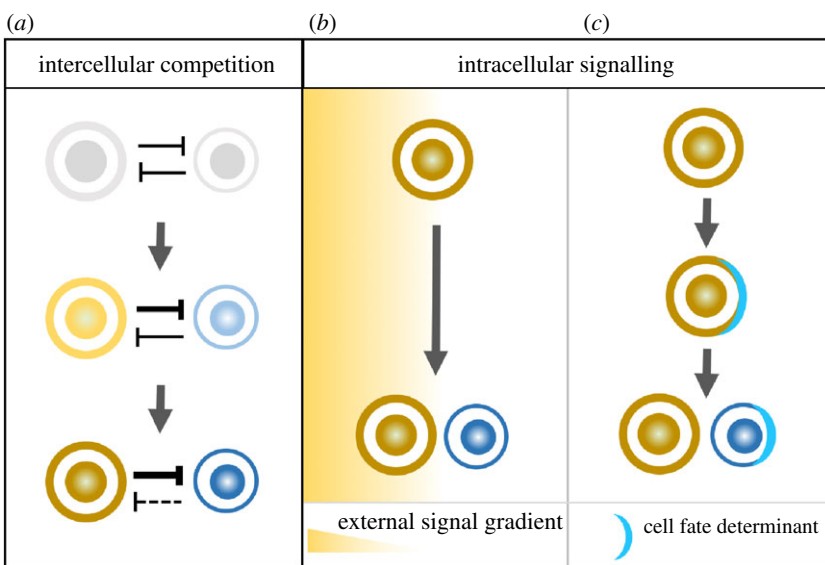

**Figure 1.** Bistability can be achieved by intercellular competition or intracellular signalling. (*a*) Lateral inhibition relies on intercellular competition between adjacent daughter cells to confer mutually exclusive binary cell fates. Small initial difference between adjacent daughter cells can be amplified and locked into distinct cell fates through intercellular feedback loops. On the other hand, (*b*) differential extracellular (such as niche-derived ligand gradient) or (*c*) intrinsic signals (such as asymmetric distribution of cell fate determinants) can dictate binary cell fate decisions via intracellular signalling network.

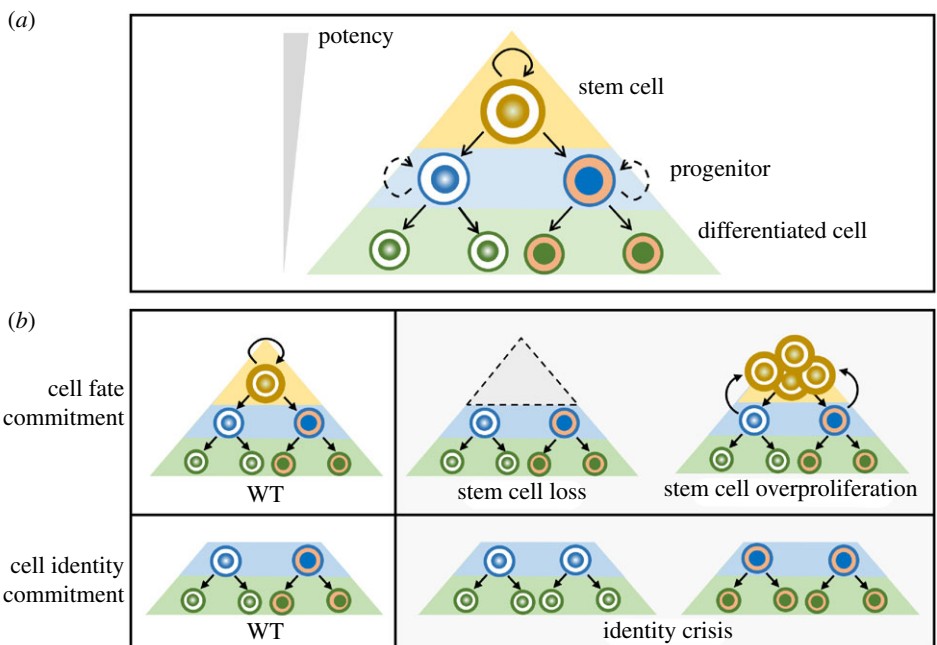

**Figure 2.** Cell fate/identity commitment in stem cell lineages. (*a*) Schematic drawing of a stem cell hierarchy. A stem cell undergoes asymmetric cell division to self-renew and produce intermediate progenitors, which in turn produce differentiated cells. Cells at the same level share similar cell fates, but acquire distinct cell identities. Here, cell fate commitment is defined as the lock-in of a cell to a specific fate within a stem cell hierarchy—a pluripotent stem cell fate, a transit-amplifying intermediate progenitor fate or a differentiated cell fate. On the other hand, cell identity commitment is defined as the acquisition of a tissue-specific, highly specialized functional potential or property. For example, cells 1 and 2 share same cell fate as terminally differentiated cells but acquire distinct functional properties and hence different cell identities. (*b*) Consequences of defective cell fate/identity commitment. Failure in cell fate or identity commitment leads to devastating consequences. Whereas defective stem cell fate commitment results in stem cell premature differentiation and tissue atrophy, defective progenitor fate commitment can lead to dedifferentiation of the progenitor back to a stem cell-like status and tumorigenesis. On the other hand, failure in cell identity commitment often results in tissue malformation in development, impairing organ functionality.

extrinsic niche signals govern the establishment of cell polarity and proper orientation of mitotic spindle, and consequently asymmetric segregation of cell fate determinants to one of the daughter cells [26–29] (figure 1). In theory, after asymmetric division of a stem cell, its daughter cells could immediately and automatically adopt distinct cell fates. However, recent time-lapse live imaging of neurogenesis in flies, fish and mice clearly demonstrated that, in reality, the establishment of an initial fate bias, or *cell fate decision*, at the end of

stem cell asymmetric division is only the beginning phase of cell fate specification [30–32]. At this stage, the fate differences between two daughter cells remain small and the cell fates still plastic. Only after the fate bias between the daughter cells becomes large enough to reach certain threshold, it can be consolidated and stabilized into distinct and irreversible cell fate outcomes. This is when the cell fates are ultimately committed and locked-in. Between cell fate decision and *cell fate commitment* (figure 2), there exists a

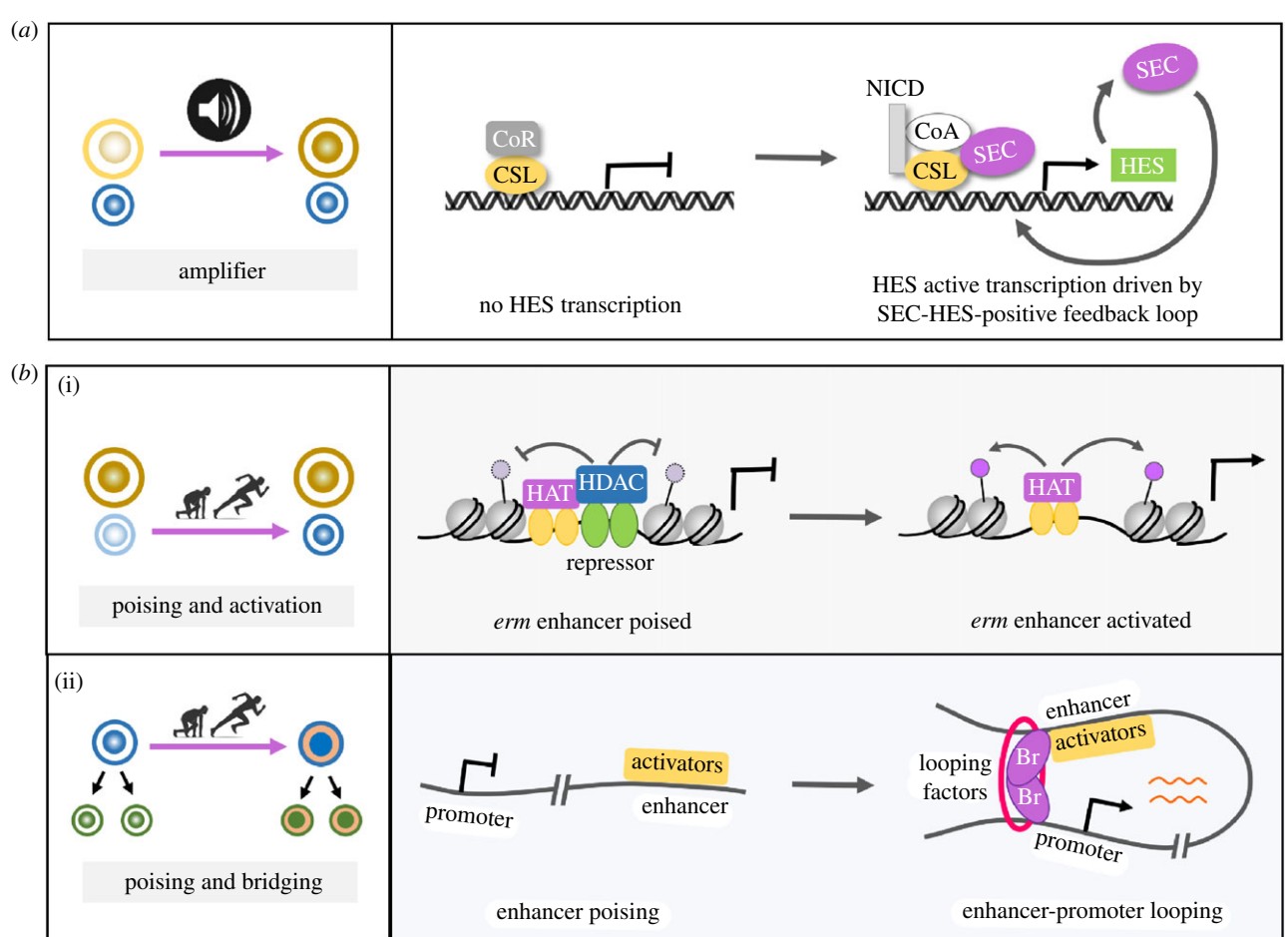

**Figure 3.** Emerging strategies ensuring timely cell fate/identity commitment. (*a*) An 'amplifier' strategy. By forming a positive feedback loop with the Notch – HES axis, SEC acts as an intrinsic amplifier to rapidly magnify and consolidate the fate bias between daughter cells and thereby drive fly type II neural stem cell fate commitment. (*b*) A 'poising' mechanism. (i) A 'poising and activation' strategy. Fast-activating enhancer of transcription factor *erm*, critical for restraining the developmental potential of neural progenitors, is poised through continual histone deacetylation in NSCs. Immediately after the birth of neural progenitors, the absence of HDAC allows rapid activation of erm transcription and hence timely fate commitment of neural progenitors. (ii) A 'poising and bridging' strategy. Spatial signal-induced transcriptional activators bind to the enhancer of cut, and a transcription factor dictates midgut – renal lineage reprogramming. The pulse of hormone ecdysone induces peak expression of temporal factor Br, which in turn acts as a transcription activator and meanwhile likely serves as a looping factor juxtaposing the distal enhancer with cut promoter, triggering timely cut transcription and midgut – renal progenitor identity conversion.

previously overlooked yet tightly controlled transition stage. Stem cells and progenitors, especially the fast-cycling ones, face the daunting challenges of ensuring timely and robust cell fate determination in every cell cycle. In these rapid-dividing cells, strategies accelerating the transition from fate decision to fate commitment need to be employed to drive cell fate determination in a speedy way.

# 2. How to speed up—emerging strategies driving timely cell fate/identity specification

## 2.1. An 'amplifier' strategy

An amplifier is a device in electronics that magnifies a small input signal to a large output signal until it reaches a desired level. It is conceivable that the fast-cycling stem cells or progenitors might ensure timely cell fate commitment through rapid amplification of the initial small fate bias upon their asymmetric division. Indeed, recent studies in fly central brain type II neural stem cell lineages revealed that the evolutionarily conserved super elongation complex (SEC), best

known for transcription elongation checkpoint control, acts as a crucial intrinsic amplifier to accelerate the previously overlooked fate transition phase and drive neural stem cell fate commitment [32,33] (figure 3). Inactivation of SEC prevents the self-reinforcing feedback loop between SEC and Notch signalling from running, resulting in NSCs with ambiguous stem cell identity and ultimate loss of stemness. Conversely, ectopic overactivation of SEC initiates and sustains this positive feedback loop within neural progenitors, driving dedifferentiation and tumorigenesis [32,33]. Given that SEC regulates rapid transcriptional induction in response to dynamic developmental or environmental cues [34,35], it is particularly suitable for being an amplifier in driving timely cell fate commitment. It is not surprising that rapidly dividing stem cells choose SEC as a regulatory component to induce immediate activation of master fate-specifying genes that in turn form a self-amplifying loop with SEC to rapidly magnify the initial fate bias and ensure prompt fate commitment. Notch signalling plays a conserved role during vertebrate embryonic neurogenesis in maintaining the undifferentiated status of NSCs [36]. Given that SEC is also highly conserved in mammals, it is interesting to speculate that a similar amplifier mechanism is used to ensure timely mammalian NSC fate commitment.

## 2.2. A 'poising' strategy

Another emerging strategy ensuring timely cell fate/identity commitment is 'enhancer-poising'. The complete process of transcriptional activation involves a few sequential steps, including the inactive, primed, pre-activated and activated transcription states [37]. To rapidly transcribe critical fate/identity genes to catch up with the developmental timing, many of these genes are kept at a pre-activated state, primed for their timely activation in response to appropriate inducing signals. Recently, this 'enhancer-poising' strategy has been nicely exemplified in fly type II neural progenitor fate commitment [38] and midgut-to-renal progenitor natural lineage reprogramming [39] (figure 3).

### 2.2.1. Poising and activation

Timely restrain of the developmental potential of neural progenitors in fly type II central brain NSC lineages depends on the rapid expression of the highly conserved transcription factor Earmuff (Erm). Recent studies revealed that the rapid transcription of *erm* in newly born neural progenitors is achieved via a 'poising and activation' mechanism [38]. A fast-activating erm enhancer is kept at a 'poised' chromatin state through continual histone deacetylation in NSCs [38]. Despite the presence of multiple histone acetyltransferases in NSCs, their activity is counteracted by the robust deacetylase activity of HDACs, effectively preventing the premature activation of Erm in NSCs [38]. Following asymmetric NSC division, the activity of HDACs is rapidly downregulated. This permits histone acetylation on *erm* enhancer and thereby rapid activation of *erm* transcription, ensuring timely fate commitment of neural progenitors [32,38,40,41] (figure 3). This fast-activating poised enhancer mechanism [42–45] might represent a general strategy that is employed by tissue-specific stem cells to initiate differentiation programmes in their newly born intermediate progenitors.

### 2.2.2. Poising and bridging

The 'enhance-poising' strategy has also been implicated in timely fly progenitor identity conversion. Recent studies unveiled a natural midgut-to-renal lineage reprogramming event during *Drosophila* metamorphosis and identified the evolutionarily conserved homeodomain protein Cut as a master switch in this process [39]. A steep Wnt/Wingless morphogen gradient intersects with a pulse of steroid hormone ecdysone to induce cut expression in a subset of midgut progenitors and reprogramme them into renal progenitors (RPs) [39]. Mechanistically, the temporal and spatial signals inducing cut transcription in future RPs intersect through a 'poising and bridging' strategy: spatial cues induce the binding of transcription activator TCF/Arm to the distal enhancer of cut, poising it for timely activation [39]. At the onset of metamorphosis, the pulse of hormone ecdysone induced peak expression of temporal factor Broad (Br). Br, in turn, acts as a transcription activator through its physical interaction with TCF/Arm. Meanwhile, Br serves as a looping factor juxtaposing the TCF/Arm-bound enhancer with cut promoter via its self-association, triggering timely cut transcription and hence midgut-renal lineage reprogramming [39] (figure 3). Importantly, because protein dimerization or oligomerization occurs only when the protein concentration rises above certain threshold [46], such protein dimerization-based regulatory mechanisms [39,47–49] are ideal for integrating and translating gradual changes in temporal and spatial signalling strength into a timely and all-or-none biological event such as cell identity switch.

## 3. Perspectives

As a slightly modified version of Lewis Wolpert's famous 'gastrulation' quote: 'It is not birth, movement, or death, but fate specification, which is truly the most important decision in a cell's life' [50]. Once thought an immediate and automatic step, cell fate/identity specification has recently been found to be a progressive and tightly regulated process. Strategies that accelerate the transition from cell fate/identity decision to commitment must be in place to ensure timely and robust fate/identity specification. Recent studies identified positive feedback loop-based 'amplifier' mechanism and 'enhancer-poising' mechanism as two such strategies used in various developmental settings. Future work using powerful model systems such as fly stem cell lineages and naturally occurring lineage reprogramming promise to unveil new regulatory principles underlying timely fate/identity determination. Advanced time-lapse live imaging technique precisely monitoring gene transcription and 3D chromatin dynamics *in vivo* will certainly be helpful in extending and deepening our understanding of the rapid cell fate/identity commitment process.

Data accessibility. This article has no additional data.

Competing interests. We declare we have no competing interests.

Funding. The work of the authors is supported by grants from the National Natural Science Foundation of China (31471372 and 31771629), Ministry of Education Key Laboratory of Cell Proliferation and Differentiation and the Peking-Tsinghua Joint Center for Life Sciences (CLS) awarded to Y.S., CLS Postdoctoral Fellowship to K.L. and Peking University Presidential Predoctoral Fellowship to K.X.

Acknowledgements. We thank members in the Song Laboratory for helpful discussions regarding the manuscript.

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
