## [Reviewer comments · Open Biology]

Review History

RSOB-18-0243.R0 (Original submission)

Review form: Reviewer 1

Recommendation

Accept with minor revision (please list in comments)

Are each of the following suitable for general readers?

- a) **Title**
Yes
- b) **Summary**
Yes
- c) **Introduction**
Yes

Is the length of the paper justified?

Yes

Should the paper be seen by a specialist statistical reviewer?

No

Is it clear how to make all supporting data available?

Not Applicable

Is the supplementary material necessary; and if so is it adequate and clear?

Not Applicable

Do you have any ethical concerns with this paper?

No

Comments to the Author

In this review, the authors discussed some recent experimental results about the cell fate or identity commitment. Without doubt, this is an interesting topic in developmental biology. In fact, since 1950s, Waddington's epigenetic landscape is probably the most famous and most powerful metaphor in developmental biology, which depicts how a cell progresses from an undifferentiated state to one of a number of discrete, distinct, differentiated cell fates during development. Waddington's theory, however, strongly implies that the determination of cell fate in the development should be considered to be a series process of dynamic bifurcation determined by some crucial biochemical reaction dynamics. For example, a recent study on the determination of *Drosophila* ovarian germline stem cell fate provided a strong experimental evidence and dynamical analysis, in which there is a feedback loop with bistable regulation induced by an external BMP signal (Curr. Biol. 22, 515 (2012)). In any case, the mechanism of bistable (or multi-stable) regulation plays an important role in determining cell fate. Based on the bistable regulation, Ferrell developed the theoretical concept of cell-fate induction (Curr. Biol. 22, R458 (2012)). Moreover, Chen et al. also pointed out that the fate differentiation of early blastomeres is not only related to the random distribution of molecular signals during cell division, but also depends on the subsequent bistable regulation mechanism (Development 142, 3468-3477 (2015)). So, I would suggest that, if it is possible, the authors include some new results about the dynamical mechanism in the determination of cell fate in this review.

Decision letter (RSOB-18-0243.R0)

18-Jan-2019

Dear Dr Song,

We are pleased to inform you that your manuscript RSOB-18-0243 entitled "Faster, Higher, Stronger: Timely and robust cell fate/identity commitment in stem cell lineages" has been accepted by the Editor for publication in Open Biology. The reviewer(s) have recommended publication, but also suggest some minor revisions to your manuscript. Therefore, we invite you to respond to the reviewer(s)' comments and revise your manuscript.

Please submit the revised version of your manuscript within 14 days. If you do not think you will be able to meet this date please let us know immediately and we can extend this deadline for you.

- 1) A text file of the manuscript (doc, txt, rtf or tex), including the references, tables (including captions) and figure captions. Please remove any tracked changes from the text before submission. PDF files are not an accepted format for the "Main Document".
- 2) A separate electronic file of each figure (tiff, EPS or print-quality PDF preferred). The format should be produced directly from original creation package, or original software format. Please note that PowerPoint files are not accepted.
- 3) Electronic supplementary material: this should be contained in a separate file from the main text and meet our ESM criteria (see <http://royalsocietypublishing.org/instructions-authors#question5>). All supplementary materials accompanying an accepted article will be treated as in their final form. They will be published alongside the paper on the journal website and posted on the online figshare repository. Files on figshare will be made available approximately one week before the accompanying article so that the supplementary material can be attributed a unique DOI.

Online supplementary material will also carry the title and description provided during submission, so please ensure these are accurate and informative. Note that the Royal Society will not edit or typeset supplementary material and it will be hosted as provided. Please ensure that the supplementary material includes the paper details (authors, title, journal name, article DOI). Your article DOI will be 10.1098/rsob.2016[*last 4 digits of e.g. 10.1098/rsob.20160049*].

- 4) A media summary: a short non-technical summary (up to 100 words) of the key findings/importance of your manuscript. Please try to write in simple English, avoid jargon, explain the importance of the topic, outline the main implications and describe why this topic is newsworthy.

Images

Sincerely,

The Open Biology Team
mailto:openbiology@royalsociety.org

Reviewer's Comments to Author:

In this review, the authors discussed some recent experimental results about the cell fate or identity commitment. Without doubt, this is an interesting topic in developmental biology.

In fact, since 1950s, Waddington's epigenetic landscape is probably the most famous and most powerful metaphor in developmental biology, which depicts how a cell progresses from an undifferentiated state to one of a number of discrete, distinct, differentiated cell fates during development. Waddington's theory, however, strongly implies that the determination of cell fate in the development should be considered to be a series process of dynamic bifurcation determined by some crucial biochemical reaction dynamics. For example, a recent study on the determination of *Drosophila* ovarian germline stem cell fate provided a strong experimental evidence and dynamical analysis, in which there is a feedback loop with bistable regulation induced by an external BMP signal (Curr. Biol. 22, 515 (2012)). In any case, the mechanism of bistable (or multi-stable) regulation plays an important role in determining cell fate. Based on the bistable regulation, Ferrell developed the theoretical concept of cell-fate induction (Curr. Biol. 22, R458 (2012)).

Moreover, Chen et al. also pointed out that the fate differentiation of early blastomeres is not only related to the random distribution of molecular signals during cell division, but also depends on the subsequent bistable regulation mechanism (Development 142, 3468-3477 (2015)). So, I would suggest that, if it is possible, the authors include some new results about the dynamical mechanism in the determination of cell fate in this review.

Author's Response to Decision Letter for (RSOB-18-0243.R0)

See Appendix A.

Decision letter (RSOB-18-0243.R1)

30-Jan-2019

Dear Dr Song,

We are pleased to inform you that your manuscript entitled "Faster, Higher, Stronger: Timely and robust cell fate/identity commitment in stem cell lineages" has been accepted by the Editor for publication in Open Biology.

Sincerely,

The Open Biology Team
mailto: openbiology@royalsociety.org

Yan Song, Ph.D.
Principal Investigator
School of Life Sciences
Peking-Tsinghua Center for Life Sciences
Peking University
Beijing, 100871, P.R.China
Tel: +86-10-6275-2120
E-mail: yan.song@pku.edu.cn

January 28, 2019

Dear Dr. Glover,

Thank you very much for your favorite decision on our manuscript RSOB-18-0243 entitled "Faster, Higher, Stronger: Timely and robust cell fate/identity commitment in stem cell lineages". We have now followed the reviewers' insightful suggestions and revised the manuscript. Please see our responses to reviewers' comments shown below.

Thank you very much for your time, effort and careful consideration of our manuscript. We sincerely hope that you find our revised and finalized manuscript appropriate for publication in *Open Biology*.

Best wishes,

Yan Song

Our response to reviewers' comments:

Reviewers' comments:

In this review, the authors discussed some recent experimental results about the cell fate or identity commitment. Without doubt, this is an interesting topic in developmental biology.

In fact, since 1950s, Waddington's epigenetic landscape is probably the most famous and most powerful metaphor in developmental biology, which depicts how a cell progresses from an undifferentiated state to one of a number of discrete, distinct, differentiated cell fates during development. Waddington's theory, however, strongly implies that the determination of cell fate in the development should be considered to be a series process of dynamic bifurcation determined by some crucial biochemical reaction dynamics. For example, a recent study on the determination of

Drosophila ovarian germline stem cell fate provided a strong experimental evidence and dynamical analysis, in which there is a feedback loop with bistable regulation induced by an external BMP signal (Curr. Biol. 22, 515 (2012)). In any case, the mechanism of bistable (or multi-stable) regulation plays an important role in determining cell fate. Based on the bistable regulation, Ferrell developed the theoretical concept of cell-fate induction (Curr. Biol. 22, R458 (2012)).

Moreover, Chen et al. also pointed out that the fate differentiation of early blastomeres is not only related to the random distribution of molecular signals during cell division, but also depends on the subsequent bistable regulation mechanism (Development 142, 3468-3477 (2015)). So, I would suggest that, if it is possible, the authors include some new results about the dynamical mechanism in the determination of cell fate in this review.

We thank reviewers for reviewing this manuscript and for providing us very insightful suggestions. We have now followed reviewers' comments and added one new paragraph with subtitle "Cell fate and bistability" (line 21- line 42) and one new figure (new Figure 1) in the "Introduction" section to discuss how bistable regulation mechanism governs cell fate determination. In this paragraph, many references, including (Curr. Biol. 22, 515 (2012); (Curr. Biol. 22, R458 (2012) and (Development 142, 3468-3477 (2015) as suggested by the reviewers, have been properly cited. The detailed text, figure and figure legend are shown below:

"Cell fate and bistability"

Cell fate decision-making in stem cell lineages is often binary: that is, newly-born sibling cells ultimately rest in two discrete, steady and switch-like equilibrium states (Balazsi et al., 2011; Bertrand and Hobert, 2010; Losick and Desplan, 2008). Such bistable cell fate specification can be achieved by either intercellular or intracellular signaling network (Brandman and Meyer, 2008; Losick and Desplan, 2008). Lateral inhibition is a typical intercellular competition that diversifies cell fates: two adjacent sibling cells inhibit each other, eventually resulting in a winner cell and a loser cell that adopt mutually exclusive cell fates (Figure 1). A classic example is Notch-Delta-mediated lateral inhibition, which has been implicated in cell fate determination within various stem cell lineages (Artavanis-Tsakonas et al., 1999; Koch et al., 2013). On the other hand, intracellular bistable regulation converts differential external or intrinsic signals into binary cell fates (Ferrell, 2012; He et al., 2019; Homem et al., 2015; Jiang and Edgar, 2012; Kohwi and Doe, 2013; Lehmann, 2012; Lin, 2008; Losick et al., 2011; Yoon et al., 2018) (Figure 1). For example, in *Drosophila* ovarian germline stem cell (GSC) lineages, differential strengths of niche-derived BMP signal can be sensed by a GSC and its sibling cystoblast cell, and converted into bistable cell fates through cell-autonomous negative feedback loops (Xia et al., 2010; Xia et al., 2012). For another instance, asymmetric segregation of Numb protein during mitosis dictate neural stem cell

versus neural progenitor/precursor binary cell fate decisions (Bowman et al., 2008; Brand and Livesey, 2011; Li et al., 2013; Lin et al., 2010; Song and Lu, 2011, 2012). Similar strategies are employed during cell fate decision-making in a wide range of stem cell lineages (Ferrell, 2012), including the very first cell fate differentiation events in early mammalian development (Shi et al., 2015; Wang et al., 2018).

The intercellular and intracellular regulatory strategies are not mutually exclusive. Instead, they are complementary to each other and can be used in combination to confer differential cell fates in a precise and robust manner.”

Figure1. Bistability can be achieved by intercellular competition or intracellular signaling. Lateral inhibition (left) relies on intercellular competition between adjacent daughter cells to confer mutually exclusive binary cell fates. Small initial difference between adjacent daughter cells can be amplified and locked into distinct cell fates through intercellular feedback loops. On the other hand, differential extracellular [such as niche-derived ligand gradient (middle)] or intrinsic signals [such as asymmetric distribution of cell fate determinants (right)] can dictate binary cell fate decisions via intracellular signaling network.